# EMERGENCE OF COMPOSITIONAL LANGUAGE WITH DEEP GENERATIONAL TRANSMISSION

## ABSTRACT

Recent work has studied the emergence of language among deep reinforcement learning agents that must collaborate to solve a task. Of particular interest are the factors that cause language to be compositional—*i.e.* express meaning by combining words which themselves have meaning. Evolutionary linguists have found that in addition to structural priors like those already studied in deep learning, the dynamics of transmitting language from generation to generation contribute significantly to the emergence of compositionality. In this paper, we introduce these cultural evolutionary dynamics into language emergence by periodically replacing agents in a population to create a knowledge gap, implicitly inducing cultural transmission of language. We show that this implicit cultural transmission encourages the resulting languages to exhibit better compositional generalization.

## 1 INTRODUCTION

Compositionality is an important structure of language that reflects a disentangled understanding of the world – enabling the expression of infinitely many concepts using finitely many elements. Agents that have compositional understandings of the world generalize in obviously correct ways even in the face of limited training examples (Lake & Baroni, 2018). For example, an agent with a compositional understanding of `blue squares` and `purple triangles` should also understand `purple squares` without directly observing any of them. Developing artificial agents that can ground, understand, and produce compositional (and therefore more interpretable) language could greatly improve generalization to new instances and ease human-AI interactions.

In building theories of how compositionality emerges in human languages, work in evolutionary linguistics looks to the process of cultural transmission (Kirby, 2001; Kirby et al., 2008). Cultural transmission of language occurs when a group of agents pass their language on to a new group of agents, *e.g.* parents who teach their children to speak as they do. Because this education is incomplete and biased, it allows the language itself to change over time via a process known as *cultural evolution*. This paradigm (Kirby et al., 2014) explains the emergence of compositionality as a result of expressivity and compressibility – *i.e.* to be most effective, a language should be expressive enough to differentiate between all possible meanings (*e.g.*, objects) and compressible enough to be learned easily. Work in the evolutionary linguistics community has shown that over multiple 'generations' these competing pressures result in the emergence of compositional languages both in simulation (Kirby, 2001) and with human subjects (Kirby et al., 2008). These studies aim to understand humans whereas we want to understand and design artificial neural networks.

Approaching the problem from another direction, recent work in AI has studied language emergence in such multi-agent, goal-driven tasks. These works have demonstrated that agent languages will emerge to enable coordination-centric tasks to be solved without direct or even indirect language supervision (Foerster et al., 2016; Sukhbaatar et al., 2016; Lazaridou et al., 2017; Das et al., 2017). However, the resulting languages are usually not compositional (Kottur et al., 2017) and are difficult to interpret, even by other machines (Andreas et al., 2017). Some existing work has studied means to encourage compositional language formation (Mordatch & Abbeel, 2018; Kottur et al., 2017), but these settings study fixed populations of agents – *i.e.* examining language within a single generation.

**In this work we bridge these two areas – examining the effect of generational cultural transmission on the compositionality of emergent languages in a multi-agent, goal-driven setting.**

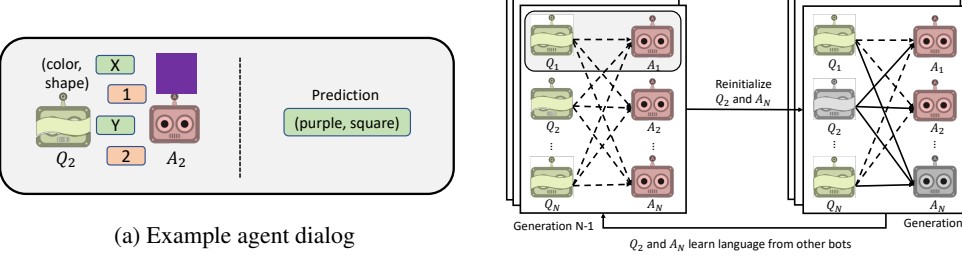

(a) Example agent dialog

(b) Implicit cultural transmission

Figure 1: We introduce cultural transmission into language emergence between neural agents. The starting point of our study is a goal-oriented dialog task (similar to that of Kottur et al. (2017)), summarized in Fig. 1a. During learning we periodically replace some agents with new ones (gray agents). These new agents do not know any language, but instead of creating one they learn it from older agents. This creates generations of language that become more compositional over time.

We study this in the context of a cooperative dialog-based reference game involving two agents communicating in discrete symbols (Kottur et al., 2017); an example dialog is shown in Fig. 1a. To examine cultural transmission, we extend this setting to a population of agents (Fig. 1b) and introduce a simple mechanism to induce the expressivity and compressibility pressures inherent in cultural transmission. Specifically, we periodically re-initialize some subset of the agents in the population. In order to perform well at the task, the population's emergent language must be sufficiently expressive to reference all the objects (expressivity) and must be easily learnable by these 'new' agents (compressibility). The new agents have a randomized language whereas the surviving agents already know a grounded language. This "knowledge gap" creates an implicit 'teaching' setting that is analogous to the explicit transmission stage in models of *iterative learning* (Kirby, 2001).

Through our experiments and analysis, we show that periodic agent replacement is an effective way to induce cultural transmission and yields more compositionally generalizable language in our setting. To summarize, our contributions are:

– We propose a method for inducing implicit cultural transmission in neural language models.
– We introduce new metrics to measure the similarity between agent languages and verify cultural transmission has occurred as a result of our periodic agent replacement protocol.
– We show our cultural transmission procedure induces compositionality in neural language models, going from 13% accuracy on a compositionally novel test set to 46% in the best configuration. Further, we show this is complementary with previous priors which encourage compositionality.

## 2 TASK & TALK: A TESTBED FOR COMPOSITIONAL LANGUAGE EMERGENCE

We consider the cooperative *Task & Talk* reference game introduced in Kottur et al. (2017). Shown in Fig. 1a, the game is played by two agents – one who observes an attributed object – *e.g.* (purple, solid, square) – and another who is given a task to retrieve a subset of these attributes over the course of the dialog – *e.g.* (color, shape). The dialog itself consists of two rounds of agents exchanging single-token utterances from fixed vocabularies. At the end of the dialog, the task-aware agent must report the requested attributes and both agents are rewarded for correct predictions. This causes a language grounded in the objects to *emerge* because there is no other way to solve the task.

A compositional solution to this task can look like a question-answer style dialog where the task-aware agent queries the other for specific attributes (Fig. 1a) – *e.g.* uttering "X" requesting the color to which the other agent replies "1" indicating purple. Importantly, this pattern would persist regardless of the other attribute values of the object (*e.g.* for all (purple, *, *) objects). However, as there is no grounding supervision provided, agents must learn to associate specific meanings to specific words and it is unlikely for compositional languages to emerge purely by chance. Given a color task, an agent might use "1" for (purple, solid, square) and then "2" for (purple, solid, circle). It is impossible for other agents to know that "2" means purple without having seen (purple, solid, circle), so compositional language is essential for generalization to compositionally novel instances.

**Models.** To formalize this setting, let Q-bot and A-bot be agent policies parameterized by neural networks $Q$ and $A$ respectively. At each round $t$, Q-bot observes the task $x_Q$ and it's memory of the dialog so far $h_Q^{t-1}$ and produces a single-token utterance $m_Q^t \in \mathcal{V}$ from the vocabulary $\mathcal{V}$. Functionally, $m_Q^t, h_Q^t = Q(m_A^{t-1}, x_Q, h_Q^{t-1})$ where $m_A^{t-1}$ is A-bot's reply in the previous round. Likewise, A-bot responds by computing $m_A^t, h_A^t = A(m_Q^t, x_A, h_A^{t-1})$ where $x_A$ is the object instance represented symbolically by concatenating 3 one-hot vectors, one per attribute. After two rounds, Q-bot must respond to the task, predicting the requested attribute pair $\hat{u} = U(x_Q, h_Q^T)$ as a function of the task and Q-bot's final memory state. Both agents are rewarded if both attributes are correct (no partial credit). We follow the neural network architectures of $Q$, $A$, and $U$ from Kottur et al. (2017).

**Measuring Compositional Generalization.** Kottur et al. (2017) generated a synthetic dataset consisting of three attribute types (color, shape, style) each with four values (*e.g.* red, blue, square, star, dotted, solid, ...) and six tasks, one task for each ordered pair of different attribute types. In total, this results in 64 unique instances and 384 task-instance pairs. To evaluate compositionality, Kottur et al. (2017) held out 12 random instances for testing. Given the closed-world set of instances, these 12 are guaranteed to be never-before-seen triplets of attribute values; however, each individual value has been seen in training in other triplets. As such, accuracy on this set is a measure of compositional generalization.

**Shortcomings of Kottur et al. (2017) Evaluation.** In our investigations, we found some shortcomings in the evaluation protocol of Kottur et al. (2017). First, the authors do not report variance over multiple runs or different random test-sets which we found to be significant. Second, the strategy of randomly selecting the test set can still reward some only partially compositional strategies. For instance, suppose agents develop a language that uses single words to refer to attribute pairs like (red, *, triangle) and (red, filled, *). Such agents might generalize to an unseen instance (red, filled, triangle) by composing the 'paired' words above instead of disentangling individual attributes.

We make two modifications to address these issues. Our results are reported as means and variances estimated from multiple training runs with different random seeds evaluated with 4-way cross-validation. We also introduce a harder dataset where instead of withholding random individual instances (e.g., (green,dotted,triangle),...) as in Kottur et al. (2017), we withhold all instances for a set of attribute pairs (e.g., (green,dotted,*),(red,solid,*),...). We will refer to datasets generated in this fashion as **novel pair** and the original dataset as **novel instance**. We report on both settings for comparison, but find our new setting to be significantly more challenging in practice – requiring a stricter notion of compositionality that is more closely aligned with human intuitions about these attributes.

## 3 COMPOSITIONAL LANGUAGE EMERGENCE WITH CULTURAL TRANSMISSION

In iterative learning models of cultural transmission from evolutionary linguistics, competing pressures towards expressivity and compressibility have been shown to induce compositionality over multiple 'generations' of language transfer (Kirby, 2001; Kirby et al., 2008). The goal-driven nature of our reference game already encourages expressivity – agents must be able to refer to the objects in order to succeed. To introduce compressibility pressure and parallel literature in evolutionary linguistics, we introduce a population of agents which regularly has members replaced by new agents that lack any understanding of the remaining population's language. As this paradigm lacks explicit teaching steps where new agents are trained to ground existing words, we consider this approach as a means of implicit cultural transmission.

**Populations of Agents.** We consider a population of Q-bots $\{Q^1, \ldots, Q^{N_Q}\}$ and a population of A-bots $\{A^1, \ldots, A^{N_A}\}$ with each agent having a different set of parameters. At each iteration during learning, we sample a random Q-bot-A-bot pair to interact and receive updates – *i.e.* the red line (2) in Alg. 1. As any Q-bot may be made to communicate with any A-bot, there is pressure for the population to adopt a unified language. Likewise, when an agent is reinitialized it will receive positive reward much more quickly when it happens to use language that its conversational partners understand. Furthermore, 'compressible' languages that are easier to learn will result in greater reward for the population in the face of periodic re-initialization of agents.

---

**Algorithm 1:** Training with Replacement and Multiple Agents

---

1 **for** *epoch* $e = 1, \ldots, N_{epochs}$ **do**
2 $\quad$ Sample Q-bot $i_Q$ from $\mathcal{U}\{1, N_Q\}$ and A-bot $i_A$ from $\mathcal{U}\{1, N_A\}$
3 $\quad$ **for** $x_Q, x_A, u$ *in each batch* **do**
4 $\quad\quad$ **for** *dialog rounds* $t = 1, \ldots T$ **do**
5 $\quad\quad\quad$ $m_Q^t, h_Q^t = Q^{i_Q}(m_A^{t-1}, x_Q, h_Q^{t-1})$
6 $\quad\quad\quad$ $m_A^t, h_A^t = A^{i_A}(m_Q^{t-1}, x_A, h_A^{t-1})$
7 $\quad\quad$ $\hat{u} = U^{i_Q}(x_Q, h_Q^T)$
8 $\quad\quad$ Policy gradient update w.r.t. *both* Q-bot and A-bot parameters
9 $\quad$ **if** $e \bmod E = 0$ **then**
10 $\quad\quad$ Sample replacement set $B$ under policy $\pi$ and re-initialize all agents in $B$
11 **return** all Q-bots and A-bots.

---

Introducing multiple agents may in itself add compressibility pressure and improve generalizations even without replacement (Raviv et al., 2018). Agents in a population have to model minor linguistic differences between conversational partners given the same memory capacity. Further, each agent provides another potential language variation that can be mimicked and perpetuated–increasing language diversity early in training. We examine these effects through no-replacement baselines, but find that generational pressure where some agents know less than others can also be important for compositionality in our setting.

**Replacement.** In order to create a notion of 'generations' we replace agents periodically. Let $\pi$ be some replacement strategy that returns a subset of the population. Every $E$ epochs, we call $\pi$ and reinitialize the parameters and optimizers for the corresponding agents (blue lines 9-10 in Alg. 1). We investigate three settings of $\pi$ (appendix A.2 for more details):

– **Uniform Random.** Sample an A-bot and Q-bot from uniform random distributions.
– **Epsilon Greedy.** With probability $1-\varepsilon$ replace the A-bot and Q-bot with the lowest validation accuracy. We use $\varepsilon = 0.2$ in our experiments.
– **Oldest.** Replace the oldest A-bot and Q-bot, breaking ties with uniform random sampling.

## 4 EXPERIMENTAL SETTING

**Experimental Setting.** We evaluate on both our modified Task & Talk dataset and the original from Kottur et al. (2017), as described in Section 2. All results are reported as means and variances computed from a total of 16 trials (four random seeds each with 4-way cross-validation). We report accuracy based on Q-bot getting both elements of the task correct – corresponding to the more restrictive "Both" setting from Kottur et al. (2017).

Kottur et al. (2017) examined a series of increasingly restrictive settings in order to study conditions under which compositionality emerges. The primary variables are whether A-bot has memory (ablated by setting $h_A^t=0$) and the vocabulary sizes $\mathcal{V}_Q$ and $\mathcal{V}_A$ for Q-bot and A-bot respectively. For comparison we also evaluate in these settings: **Minimal Vocab** ( $V_Q=3$, $V_A=4$). **Memoryless + Minimal Vocab** ($V_Q=3$, $V_A=4$, $h_A^t=0$), **Overcomplete** ($V_Q=V_A=64$). We also introduce **Memoryless + Overcomplete** ($V_Q=V_A=64$, $h_A^t=0$) to complete the cross product of settings and examine the role of memory restriction in overcomplete vocabularies.

The Memoryless + Minimal Vocabulary setting results in the best compositional generalization; however, this is an extreme setting – requiring not only that the minimum number of groundable symbols be known but also that A-bot not be able to remember it's previous utterance. While we do report these settings and see quite large performance gains due to cultural transmission, we are mainly interested in the more realistic Overcomplete setting where a large pool of possible tokens is provided and both dialog agents remember the conversation.

**Model and Training Details.** Our A-bots and Q-bots have the same architectur as in Kottur et al. (2017). All agents are trained with $E = 25000$, a batch size of 1000, [1] and the Adam (Kingma & Ba, 2015) optimizer (one per bot) with learning rate 0.01. In the Multi Agent setting we use

---

[1]There is 1 batch per epoch because there are only 384 instances (64 objects $\times$ 6 tasks).

Figure 2: Test set accuracies (with standard deviations) are reported against our new harder dataset using models similar to those in Kottur et al. (2017). Our variations on cultural transmission (darker blue bars) outperform the baselines where language does not change over generations.

$N_A = N_Q = 5$. We stop training after 8 generations (199000 epochs Multi Agent; 39000 epochs Single Agent). This differs from Kottur et al. (2017), which stopped once train accuracy reached 100%. Further, we do not perform negative mining.

**Baselines.** We consider a set of baseline setting to isolate the effect of our approach.

– **Single Agent Populations.** We ablate the effect of multi-agent populations by training individual A-bot-Q-bot pairs (*i.e.* populations with $N_A = N_Q = 1$). We apply the *uniform random* (either A-bot or Q-bot at random) and *oldest* (alternating between A-bot and Q-bot) replacement strategies to these agents; however, the *epsilon greedy* strategy is not well-defined here. In this setting we decrease $E$ from 25000 to 5000 to keep the average number of gradient updates for each agent constant with respect to the multi-agent experiments.

– **No Replacement.** We also consider the effect of replacing no agents at all, but still allowing the agents to train for the full 199,000 (39,000) epochs. Improvement over this baseline shows the gains from our replacement strategy under identical computational budgets.

## 5 RESULTS AND ANALYSIS

### 5.1 IMPACT OF CULTURAL TRANSMISSION ON COMPOSITIONAL GENERALIZATION

Results with standard deviations against our harder dataset are reported in Fig. 2. We compared methods and models using dependent paired t-tests and reported the resulting p-values in Section A.3. Result on the original Task & Talk dataset are in Section A.1.

**Cultural transmission induces compositionality.** Our main result is that cultural transmission approaches outperform baselines without cultural transmission. This can be seen by noting that for each model type in Fig. 2, the 3 darker blue bars (Multi Agent Replacement approaches) are largest. After running a dependent paired t-test against all pairs of baselines and cultural transmission approaches we find a meaningful difference in all cases ($p \leq 0.05$). This is strong support for our claim that our version of cultural transmission encourages compositional language because it causes better generalization to novel compositions of attributes.

Next we go on to discuss some additional trends we hope the community will find useful.

**Population dynamics without replacement usually lead to some compositionality.** The *Multi Agent No Replacement* policies usually outperform than the *Single Agent No Replacement* policies, though the difference isn't very significant in the except in the *Overcomplete* and *Minimal Vocab* settings. This agrees with recent work from evolutionary linguistics, where multiple agents can lead to compositionality without generational transmission Raviv et al. (2018).

**Variations in replacement strategy tend to not affect performance.** The *Multi Agent Uniform Random/Epsilon Greedy/Oldest* replacement strategies are not largely or consistently different from one another across model variations. This suggests that while some agent replacement needs to occur, it is not critical whether agents with worse language are replaced or whether there is a pool of similarly typed agents to remember knowledge lost from older generations. The main factor is that new agents learn in the presence of others who already know a language.

**Cultural transmission is complementary with other factors that encourage compositionality.** As in Kottur et al. (2017), we find the *Memoryless + Small Vocab* model is clearly the best. This agrees with factors noted elsewhere Kottur et al. (2017); Mordatch & Abbeel (2018); Nowak et al. (2000) and shows how many different factors can affect the emergence of compositionality.

**Removing memory makes only minor differences.** Removing memory makes no difference (negative or positive) in Single Agent settings, but it can have a relatively small effect in Multi Agent settings, helping *Small Vocab* models and hurting *Overcomplete* models. While our approach is complementary with minimizing vocab size to increase compositionality, its makes memory removal less useful. As the *Memoryless + Overcomplete* setting has not been reported before, these results suggest that the relationship between inter-round memory and compositionality is not clear.

Overall, these results show that adding cultural transmission to neural dialog agents improves the compositional generalization of the languages learned by those agents in a way complementary to other priors. It thereby shows how to transfer the cultural transmission principle from evolutionary linguistics to deep learning.

## 5.2 IS GENERATIONAL TRANSMISSION OCCURRING?

Because it is implicit, cultural transmission may not actually be occurring; improvements may be from other sources. How can we measure cultural transmission? We focus on A-bots and take a simple approach. We assume that if two A-bots 'speak the same language' then that language was culturally transmitted. There is a combinatorial explosion of possible languages that could refer to all the objects of interest, so if the words that refer to the same object for two agents are the same then they were very likely transmitted from the other agents, rather than similar languages emerging from scratch just by chance. This leads to a simple approach: consider pairs of bots and see if they say similar things in the same context. If they do, then their language was likely transmitted.

More formally, consider the distribution of tokens A-bot $A^i$ might use to describe its object $x_A$ when talking to Q-bot $Q^k$: $p_{k,i}(m_A^t|x_A)$ or $p_{k,i}$ for short. We want to know how similar $A^i$'s language is to that of another A-bot $A^j$. We'll start by comparing those two distributions by computing the KL divergence between them and then taking an average over context (objects, Q-bots, and dialog rounds) to get our pairwise agent language similarity metric $D_{ij}$:

$$D_{ij} = \hat{E}_{x_A,k,t}\left[D_{KL}\left(p_{k,i}(m_A^t|x_A), p_{k,j}(m_A^t|x_A)\right)\right] \qquad (1)$$

Taking another average, this time over all pairs of bots (and also random seeds and cross-val folds), gives our final measure of language similarity reported in Fig. 3.

$$D = \hat{E}_{i,j \text{ s.t. } i \neq j}\left[D_{ij}\right] \qquad (2)$$

$D$ is smaller the more similar language is between bots. Note that even though $D_{ij}$ is not symmetric (because KL divergence is not), $D$ is symmetric because it averages over both directions of pairs.

We compute $D$ by sampling an empirical distribution over all messages and observations, taking 10 sample dialogues in each possible test state $(x_A, x_Q)$ of the world using the final populations of agents as in Fig. 2. Note that this metric applies to a group of agents, so we measure it for only the *Multi Agent* settings, including two new baselines colored red in Fig. 3. The *Single Agents Combined* baseline trains 4 *Single Agent No Replacement* models independently then puts them together and computes $D$ for that group. These agents only speak similar languages by chance, so $D$ is high. The *Random Initialization* baseline evaluates language similarity using newly initialized models. These agents have about a uniform distribution over words at every utterance, so their languages are both very similar and useless. For each model these baselines act like practical (not strict) upper and lower bounds on $D$, respectively.

Fig. 3 shows this language dissimilarity metric for all our settings. As we expect, the paired *Single Agents* are highly dissimilar compared to agents from *Multi Agent* populations. Further, all the replacement strategies result in increased language similarity—although the degree of this effect seems dependent on vocabulary setting. This provides some evidence that cultural transmission is occurring in *Multi Agent* settings and is encouraged by the replacement strategy in our approach. While all *Multi Agent* settings resulted in language transmission, our replacement strategies results in more compositional languages due to repeated teaching of new generations of agents.

## 5.3 VISUALIZING EMERGENT LANGUAGES

In this section we visualize the language learned by agents at various stages of training to reinforce our previous conclusions and build intuition. Each of the three sub-figures in Fig. 4 summarizes all of the conversations between a particular pair of bots for the (shape, color) task. To see how these summaries work, consider Fig. 4a. That sub-figure is divided into a $4 \times 4$ grid with 4 elements

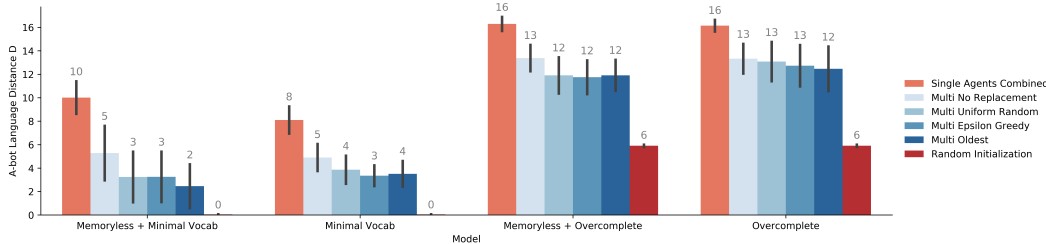

Figure 3: Do bots in a population learn similar languages? On the y-axis (eq. (2)) lower values indicate similar language. Populations evolved with our method speak similar languages, but independently evolved agents do not. Thus our implicit procedure induces cultural transmission.

in each cell, so there is one element for each of the 64 possible objects. For this task, objects in each row of the $4 \times 4$ grid have the same shape and objects in each column have the same color. To the right of each object are the two tokens A-bot used to respond to Q-bot in the two dialog rounds. Ideally they should indicate the color and shape of the object. Finally, the check-marks or Xs to the right of A-bot's utterances indicate whether Q-bot guessed correctly.

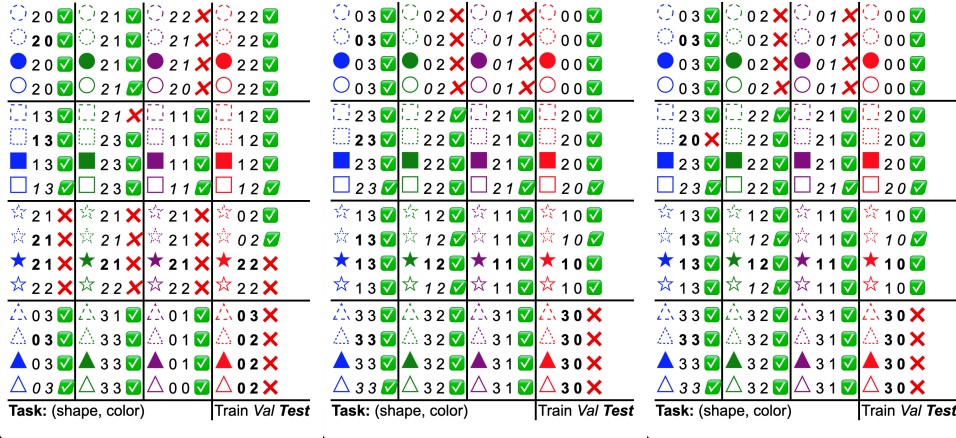

(a) Gen 1 (Single) - New A-bot    (b) Gen 8 (Multi) - New A-bot    (c) Gen 8 (Multi) - Old A-bot

Figure 4: Each sub-figure summarizes an A-bot's language, as described in Section 5.3. By comparing the baseline of Fig. 4a to a similar pair of bots from our approach Section 4b we can see that our approach encourages compositional language to emerge. Furthermore, the similarity between Fig. 4b and Fig. 4c suggests language is indeed transmitted in our approach.

From left to right: Fig. 4a summarizes the single pair from a Single Agent No Replacement run (3000 iterations old); Fig. 4b summarizes dialogs between an old Q-bot (about 23000 iterations) and a recently re-initialized A-bot (about 3000 iterations) at the 8th and final generation of a Multi Oldest run; Fig. 4c summarizes dialogs between the same old Q-bot as in Fig. 4b and an old A-bot (13000 iterations) from the same Multi Oldest experiment. Even though the A-bots in Fig. 4a and Fig. 4b have trained for about[2] the same number of iterations, the A-bot trained in the presence of other bots which already know a functional language has already learned a somewhat compositional language whereas the Single Agent A-bot has not. Furthermore, by comparing the old A-bot's language Fig. 4c with the new one Fig. 4b we can see that they are extremely similar. They even lead to the same mistakes. This again suggests that language is transmitted between bots, in agreement with our previous experiments.

## 6 RELATED WORK

**Language Evolution Causes Structure.** Researchers have spent decades studying how unique properties of human language like compositionality could have emerged. There is general agreement that people acquire language using a combination of innate cognitive capacity and learning from other language speakers (cultural transmission), with the degree of each being widely disputed Perfors (2002); Pinker & Bloom (1990). Both innate cognitive capacity and specific modern

---

[2]Due to the stochastic nature of our Multi Agent approach.

human languages like English co-evolved Briscoe (2000) via biological Pinker & Bloom (1990) and cultural Tomasello (1999); Smith (2006) evolution, respectively.

In particular, explanations of how the cultural evolution of languages could cause structure like compositionality are in abundance Nowak & Krakauer (1999); Nowak et al. (2000); Smith et al. (2003); Brighton (2002); Vogt (2005); Kirby et al. (2014); Spike et al. (2017). An important piece of the explanation of linguistic structure is the iterated learning model Kirby et al. (2014); Kirby (2001); Kirby et al. (2008) used to motivate our approach. Indeed it shows that cultural transmission causes structure in computational Kirby (2001; 2002); Christiansen & Kirby (2003); Smith et al. (2003) and human Kirby et al. (2008); Cornish et al. (2009); Scott-Phillips & Kirby (2010) experiments. Even though cultural transmission may aid the emergence of compositionality, recent results in evolutionary linguistics Raviv et al. (2018) and deep learning Kottur et al. (2017); Mordatch & Abbeel (2018) also emphasize other factors.

While existing work in deep learning has focused on biases that encourage compositionality, it has not considered settings where language is permitted to evolve over generations of agents. We have shown such an approach is viable and even complementary with other approaches.

**Language Emergence in Deep Learning.**   Recent work in deep learning has increasingly focused on multi-agent environments where deep agents learn to accomplish goals (possibly cooperative or competitive) by interacting appropriately with the environment and each other. Some of this work has shown that deep agents will develop their own language where none exists initially if driven by a task which requires communication Foerster et al. (2016); Sukhbaatar et al. (2016); Lazaridou et al. (2017). Most relevant is work which focuses on conditions under which *compositional* language emerges as deep agents learn to cooperate Mordatch & Abbeel (2018); Kottur et al. (2017). Both Mordatch & Abbeel (2018) and Kottur et al. (2017) find that limiting the vocabulary size so that there aren't too many more words than there are objects to refer to encourages compositionality, which follows earlier results in evolutionary linguistics Nowak et al. (2000). Follow up work has continued to investigate the emergence of compositional language among neural agents, mainly focusing on perceptual as opposed to symbolic input and how the structure of the input relates to the tendency for compositional language to emerge Choi et al. (2018); Havrylov & Titov (2017); Lazaridou et al. (2018). Other work has shown that Multi Agent interaction leads to better emergent translation Lee et al. (2018), but it does not measure compositionality.

**Cultural Evolution and Neural Nets.**   Somewhat recently, Bengio (2012) suggested that culturally transmitted ideas may help in escaping from local minima. Experiments in Gülçehre & Bengio (2016) support this idea by showing that supervision of intermediate representations allows a more complex toy task to be learned. Unlike our work, these experiments use direct supervision provided by the designed environment rather than indirect and implicit supervision provided by other agents.

Two concurrent works examine the role of periodic agent replacement on language emergence – albeit in different environments. In Li & Bowling (2019) replacement is used to encourage languages to be easy to teach, and this in turn causes compositionality. In Dagan et al. (2019) neural language is transmitted through a bottleneck caused by replacement. The resulting language has increased efficiency and effectiveness, with further results showing that co-evolving the agents themselves with the language amplifies the effect. Both of these works support our central observations.

## 7  CONCLUSION

In this work we investigated cultural transmission in deep neural dialog agents, applying it to language emergence. The evolutionary linguistics community has long used cultural transmission to explain how compositional languages could have emerged. The deep learning community, having recently become interested in language emergence, has not investigated that link until now. Instead of explicit models of cultural transmission familiar in evolutionary linguistics, we favor an implicit model where language is transmitted from generation to generation only because it helps agents achieve their goals. We show that this does indeed cause cultural transmission and compositionality.

**Future work.**   While our work used an implicit version of cultural transmission, we are interested in investigating the effect of explicit versions of cultural transmission on language structure. In another direction, cultural transmission may also provide an appropriate prior for neural representations of non-language information.

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

# A APPENDIX

## A.1 RESULTS ON SINGLE HELD OUT ATTRIBUTE DATASET OF KOTTUR ET AL. (2017)

In Section 4 we proposed a new harder compositional dataset different from the one in Kottur et al. (2017). For comparison, in this section we train and evaluate our models on the original dataset from Kottur et al. (2017) to show that our approach also improves still improves compositionality in this setting and to show that our new dataset is indeed harder.

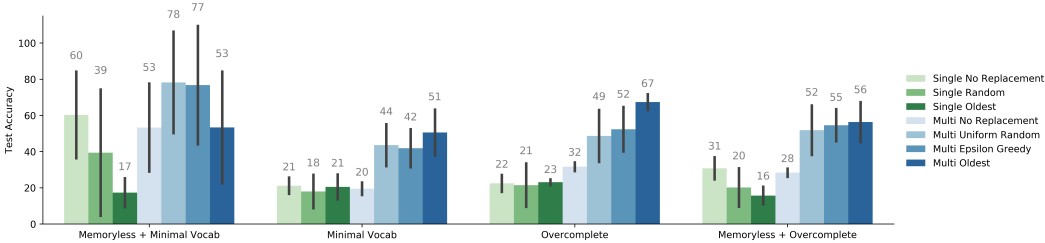

Figure 5: Test set accuracies (with standard deviations) are reported by training and evaluating the same models as in our main results Fig. 2 against the dataset from Kottur et al. (2017). These results do not perform cross-validation, following Kottur et al. (2017). They only vary across 4 different random seeds. This dataset is significantly easier than our new dataset, as indicated by the Our proposed approach still outperforms models without replacement or multiple agents.

## A.2 REPLACEMENT STRATEGIES

Our approach to cultural transmission periodically replaces agents by re-initializing them. The approach section outlines various replacement strategies (policy $\pi$), but does not detail their implementation. We do so here.

These strategies depend on a number of possible inputs:

- $e$ the current epoch
- $E$ the period of agent replacement
- $v_i^Q/v_i^A$ the validation accuracy of agent $i$ for Q-bots/A-bots. For Q-bots this is averaged over all potential A-bot partners, and vice-versa for A-bots.
- $a_i^Q/a_i^A$ the age in epochs of agent $i$ for Q-bots/A-bots

Single Agent strategies are given in Alg. 2 and Alg. 3. Multi Agent strategies are given in Alg. 4, Alg. 5, and Alg. 6. Note that Single Agent strategies always replace one agent while Multi Agent strategies always replace one Q-bot and one A-bot. An additional Replace All baseline strategy is given in Alg. **??** and generalizes to both Single and Multi Agent cases.

---

**Algorithm 2:** Single Agent - Random Replacement

1   $d \sim \mathcal{U}\{0, 1\}$
2   **if** $d = 0$ **then**
3     **return** { A-bot }
4   **else**
5     **return** { Q-bot }

---

## A.3 DETAILED RESULTS

In our experiments we compare models and we compare replacement strategies. We ran dependent paired t-tests across random seeds, cross-val folds, and replacement strategies to compare models. We ran dependent paired t-tests across random seeds, cross-val folds, and models to compare replacement strategies. The p-values for all of these t-tests are reported here.

Replacement strategy comparisons are in Fig. 7 (Single Agent) and Fig. 8 (Multi Agent). Model comparisons are in Fig. 6.

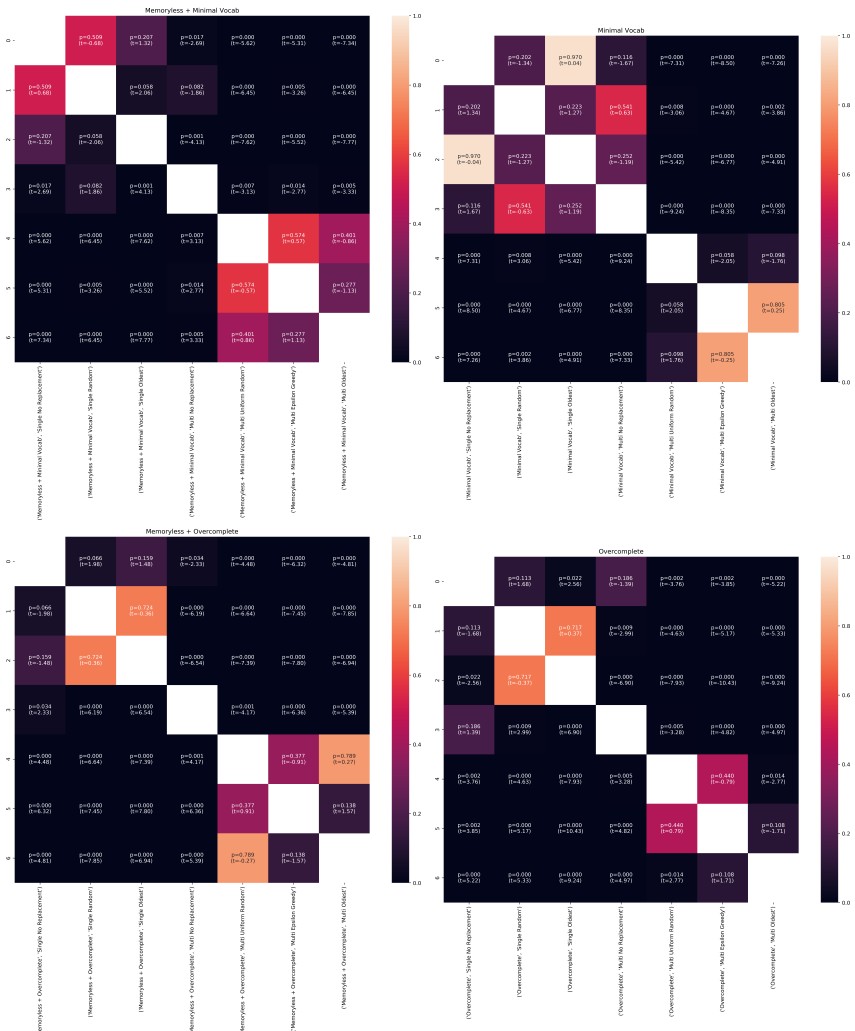

Figure 6: Replacement strategy comparison p-values.

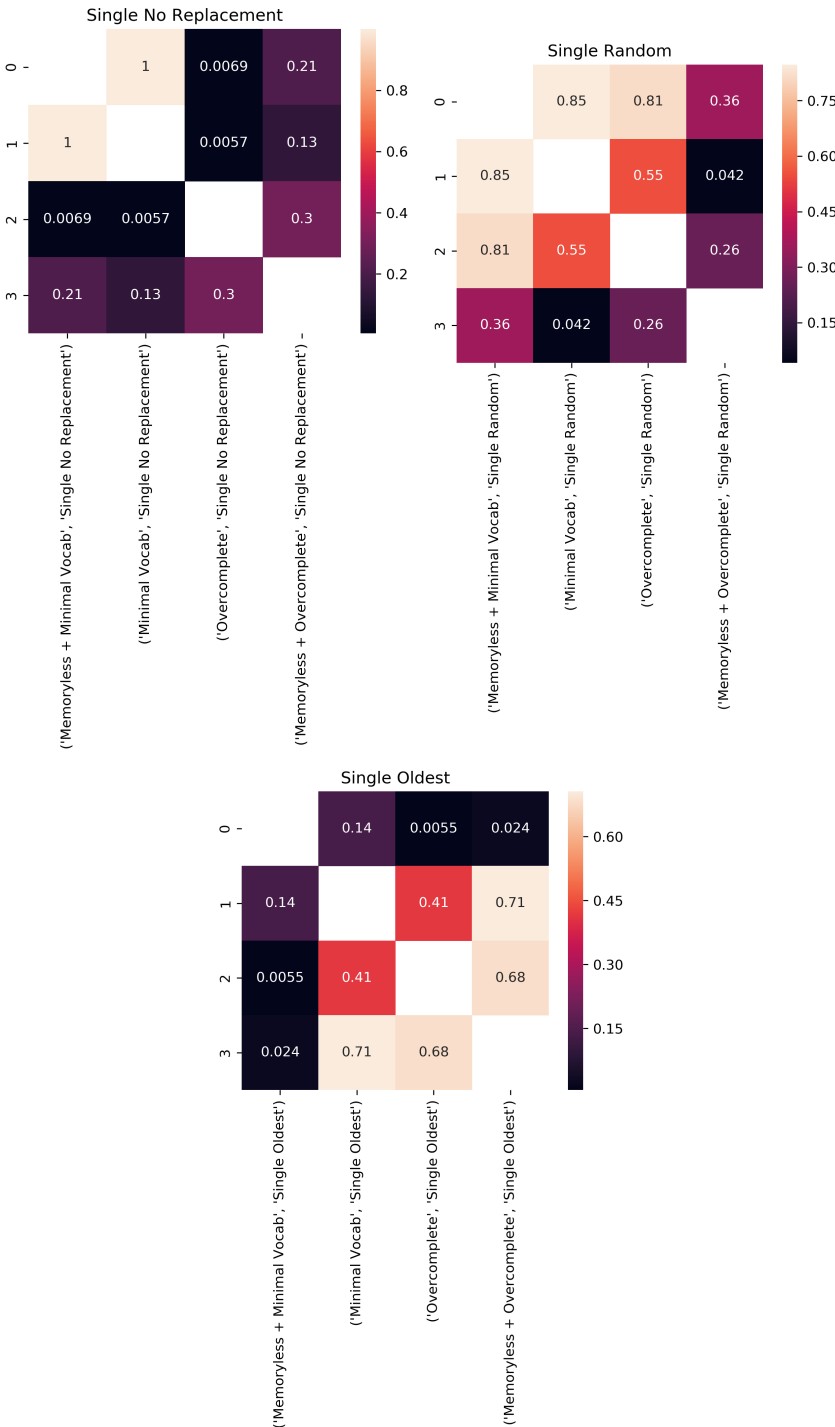

Figure 7: Single Agent model comparison p-values.

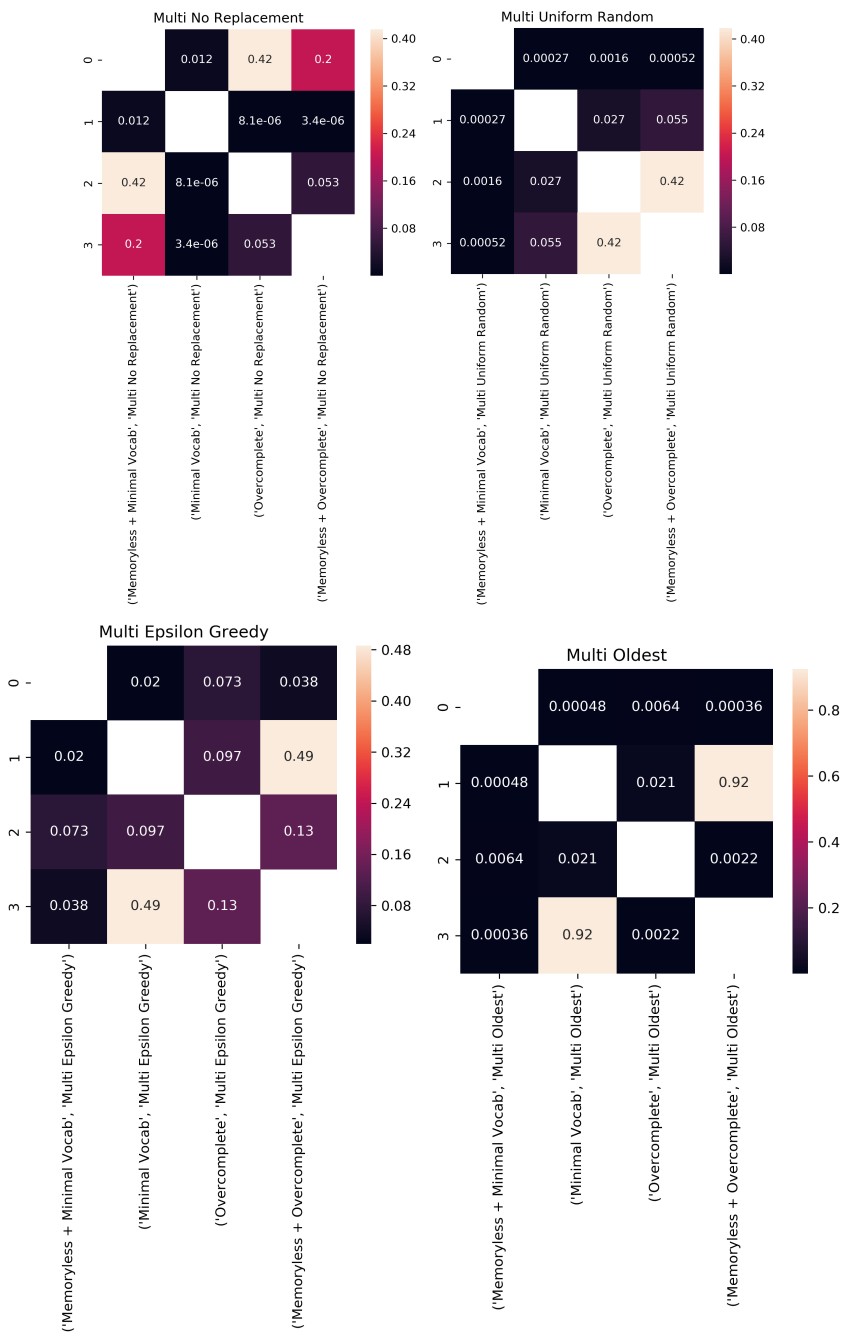

Figure 8: Multi Agent model comparison p-values.

---

**Algorithm 3:** Single Agent - Alternate Replacement

---

**1** **Input:** $e$
**2** **if** $\lfloor e/E \rfloor = 0$ **then**
**3** $\quad$ **return** { A-bot }
**4** **else**
**5** $\quad$ **return** { Q-bot }

---

**Algorithm 4:** Multi Agent - Uniform Random Replacement

---

**1** $i_A \sim \mathcal{U}\{1, N_A\}$
**2** $i_Q \sim \mathcal{U}\{1, N_Q\}$
**3** **return** { A-bot $i_A$, Q-bot $i_Q$ }

---

**Algorithm 5:** Multi Agent - Epsilon Greedy Replacement

---

**1** **Input:** $v_i^Q \forall i$, $v_i^A \forall i$, $\varepsilon \in [0, 1)$ (usually 0.2)
**2** $d \sim \mathcal{U}[0, 1)$
**3** **if** $d < \varepsilon$ **then**
**4** $\quad$ $i_A \sim \mathcal{U}\{1, N_A\}$
**5** $\quad$ $i_Q \sim \mathcal{U}\{1, N_Q\}$
**6** **else**
**7** $\quad$ $i_A = \operatorname{argmin}_i v_i^A$ (unique in our experiments)
**8** $\quad$ $i_Q = \operatorname{argmin}_i v_i^Q$ (unique in our experiments)
**9** **return** { A-bot $i_A$, Q-bot $i_Q$ }

---

**Algorithm 6:** Multi Agent - Oldest Replacement

---

**1** **Input:** $a_i^Q \forall i$, $a_i^V \forall i$
**2** $i_A = \mathcal{U}\{\operatorname{argmax}_i a_i^A\}$
**3** $i_Q = \mathcal{U}\{\operatorname{argmax}_i a_i^Q\}$
**4** **return** { A-bot $i_A$, Q-bot $i_Q$ }

---

