# OpenReview forum: "Emergence of Compositional Language with Deep Generational Transmission"
_ICLR.cc/2020/Conference — Reject_

### Official Review · AnonReviewer1 · 2019-10-25
**Official Blind Review #1**

**Rating:** 1

**Review:**

The paper continues the line of work on emergent compositionality in dialogs, and here it is extended to handle groups of interacting agents that pass language in an evolutionary way from one generation to another. The key idea is that with group of interacting agents, if some agents are replaced with new ones, then the newbies would learn the same language as the group.

General assessment:

The setup of the paper is is interesting, and the paper covers a lot of ground. The paper makes bold claims like "cultural transmission induces compositionality" and "Variations in replacement strategy tend to not affect performance". Unfortunately, the paper does not systematic provide experimental evidence to adequately support its claims.

In my experience, the emergence of compositionality (or lack of) in the setup learned here is very sensitive to various aspects of the learning setup, hence are hard to reproduce. Specifically, they the task-and-talk paper by Kottur et al may yield different conclusions, if parameters of the original experiments are modified, even slightly. The current paper does improve the evaluation protocol of Kottur 2017 by reporting variance over runs, but does not explore the parameter space more systematically, hence I am concerned that it may suffer from similar fragility.

In this light, significantly more evidence should be provided to convince that the experimental results are stable and can be reproduced.  First, one needs to show the experiments repeated over the full range of setup parameters. Including, the vocabulary sizes V_Q and V_A, the number of tasks, the number of attributes per task, and number of agents etc. Similarly, the current paper introduces a new compositional split, generated in one specific way. The effect of the split on what aspects of language emerge should be studied systematically, instead of using one "hard" and one random split.
The "evolutionary" part has a similar issue. The paper draws conclusions from three anecdotal rules for replacing the population. There is no systematic analysis of the "evolution" process, not even studying a range of the parameter epsilon, which is set 0.8 (arbitrarily? to fit the story? we do not know).  Drawing conclusions based on anecdotal evidence is bad scientific practice, that ICLR should discourage.

While the ideas in this paper are innovative and exciting, the paper promises much more than its analysis supports, and the paper is not ready for publication.

Other comments:
-- The paper states that "darker blue bars" in figure 2 are higher. The statistical analysis is not well explained, not even in the supplemental, so it is hard to tell which differences are significant. If data is paired, it would be useful to view data as a scatter plot, instead of a barplot which hides the pairing. BTW, p<0.05 is not a "strong support", but rather is the most permissive threshold. The results in the supplemental may be stronger.
-- "Variations in replacement strategy tend to not affect performance." This is a key result of the paper and mush be quantified and analyzed. Authors should define some space of replacements strategies (e.g. in in parametric ways like  how often and how many agents are replaced), then compute performance difference as a function of grid search over the parameter space and show a figure.
-- "We stop after 8 generations". Justify with data.
-- Other parts of the paper make additional claims, that should similarly be systematically analyzed and supported with data-driven evidence.


**Experience Assessment:**

I have published in this field for several years.

**Review Assessment: Checking Correctness Of Derivations And Theory:**

N/A

**Review Assessment: Checking Correctness Of Experiments:**

I carefully checked the experiments.

**Review Assessment: Thoroughness In Paper Reading:**

I read the paper at least twice and used my best judgement in assessing the paper.

---

> ### Author Response · Authors · 2019-11-14
> **Responses to R1**
>
> Thanks for the review. It raises some serious concerns, however we think they can be appropriately addressed and have attempted to do so in our responses below.
>
> __Comment (insufficient experiments)__: “In my experience, the emergence of compositionality (or lack of) in the setup learned here is very sensitive[...]” “The current paper does improve the evaluation protocol[...] but does not explore the parameter space more systematically, hence I am concerned that it may suffer from similar fragility.”
>
> “In this light, significantly more evidence should be provided to convince that the experimental results are stable and can be reproduced.”
>
> “The "evolutionary" part has a similar issue. The paper draws conclusions from three anecdotal rules for replacing the population. There is no systematic analysis of the "evolution" process[…] Drawing conclusions based on anecdotal evidence is bad scientific practice, that ICLR should discourage.”
>
> “"Variations in replacement strategy tend to not affect performance." This is a key result of the paper and mush be quantified and analyzed.”
>
> __Response__:
> We are also concerned that our experiments may suffer from some fragility, partially because of the fragility we found in the experiments of Kottur et. al. That is exactly why we extended our experiments to capture much of this variance, as mentioned. We further agree that our claims should be softer, perhaps changing "cultural transmission induces compositionality" to “our replacement strategies increase compositionality.”
>
> However, as R3 puts it, in papers like this there are “never-ending experiments” and the question is whether or not “this paper [has] enough of these never-ending experiments.”
> We agree with R3 that this paper does have enough of these never ending experiments. We think Figure 2 and the p values reported in A.3 show a clear increase in compositionality as a result of our replacement strategies and we systematically consider variations in Memory, Vocab Size, Number of Agents, and Replacement Strategy:
>
> Grid search dimensions
> (Model) Memory: Memory vs No Memory
> (Model) Vocab Size: Small vs Large/Overcomplete Vocab
> (Method) Number of Agents: Single vs Multiple Agents
> (Method) Replacement Strategy: No Replacement vs Random vs Epsilon Greedy vs Oldest
>
> We did search over how often our agents are replaced in our initial experiments and found that replacement every 25000 epochs is a good tradeoff between allowing agents to converge in a generation and enabling us to run multiple generations. New agents usually converge in this length of time. Increasing this parameter is expensive because it must be increased for every generation, making each increase 8x more expensive (given our 8 generations).
>
> We’d also like to point out that just the experiments we reported in the paper took over 3 GPU months to run. That’s a conservative back of the envelope calculation that assumes each single agent model takes 20min to train and each multi-agent model takes 6hr40min to train (about right for our implementation). This does not include the iterations our models went through before converging on the models presented in the paper, which likely multiply that figure by at least 2x or 3x. Running the additional experiments would take a long time because they would have to include all of the hyperparameter variations we have already considered.
>
> Both R4 (“the experimental results largely confirm the hypothesis”) and R3 (“The claim is clear, the hypotheses are well-stated, and the experiments look solid”) agree that our experiments support our main claim. Is our main claim invalid without the additional experiments requested here?
>
>
> __Comment__: “The statistical analysis is not well explained, not even in the supplemental, so it is hard to tell which differences are significant. If data is paired, it would be useful to view data as a scatter plot, instead of a barplot which hides the pairing.”
> __Response__: We pair the 16 different random seed/split variations. We report dependent paired t-test p values between each pair of experiments (114 tests with 16 instances in each case) in appendix A.3. We’re not sure we quite understand the plotting suggestion here. Should we plot model performance with random seed and split values on the x and y axes?
>
> __Comment__: “BTW, p<0.05 is not a "strong support", but rather is the most permissive threshold. The results in the supplemental may be stronger.”
> __Response__: In almost every case p<0.01 (comparing Replacement and No Replacement models), as reported in A.3.
>
>
> __Comment__: “Other parts of the paper make additional claims, that should similarly be systematically analyzed and supported with data-driven evidence.”
> __Response__: Could the reviewer point out the specific parts of the paper that make additional claims without evidence?

---

### Official Review · AnonReviewer3 · 2019-10-25
**Official Blind Review #3**

**Rating:** 6

**Review:**

This paper studies whether language composition may emerge by partially re-sampling new agents inside a pool of language agents. They set up a consistent experimental setting for assessing compositionality,  assess different agent architectures, e.g., memory vs. memoryless agents,  and explore how the language remains close to each other by re-sampling new agents.

The paper is well-motivated with substantial background literature on the cognitive science and emergent communication side. The claim is clear, the hypotheses are well-stated, and the experiments look solid (I particularly appreciated the paragraph on shortcoming evaluation). In the end, I enjoy reading the paper despite its density, and I could see that the authors made quite some effort in that direction.

Improvement direction, questions:
 - The authors made were careful not to take ownership of Kottur et al. 's works. Yet, the writing sometimes gives the feeling that their work is solely an extension of Kottur's work, which is not the case. It also gives the feeling that Kottur et al. is the only valid experimental setting, which is not the case (at the authors pointed out in the related work section). Thus, I would recommend to summarise at some point the similarity/difference between the two papers, or at least stop referring the paper every two lines!
 - Replacement strategy: the authors use simple replacement strategies, and conclude that it has little impact. Althought It sounds reasonable in the current setting, the conclusion may be a bit premature. I would recommend to discuss further this result with complementary experiments could be the following: see the impact of epsilon, trying tournament strategies, why 8 populations (this sounds a bit arbitrary too). I would also like to put those observations in perspective with the evolutionary literature [1], and even provide a full paragraph in the related work section.
 - Population dynamics: I am missing a key element in the paper: an analysis of the population dynamics. Although the paper deals with generational transmissions, there are no experiments that analyze the evolution of language generations after generations. Most of the experiments deal with the final convergence state. Again, I would recommend having a look at the evolutionary literature to see which protocol they use to analyze such behavior.
 - Literature side: The authors did an excellent job on the emergent communication and cognitive science side. I think that it is worth extending the comparison further. For instance:
    * generational transmission can be studied in the light of game theory [2] where compositionality can be seen as a Nash Equilibrium between agent.
    * generational transmission is a form of dynamic distillation [3]
    * and evolutionary algorithms!
 - I had some difficulties in understanding Fig4, and the final-take away correctly. Would it be possible to give me one or two examples to correctly parse the table? More generally, I would recommend to add a few lines with some concrete and cherry-picked examples from the experiments to help the reader to have more intuition).
 - In a similar spirit, it is hard to interpret the distance in Figure 3. What would correspond to an increase of 1pt of distance? Having said that, the experiment is sound, and it is insightful.
 - reproducible: having a final table in the array in the appendix could be very helpful
 - crazy experiment: even if I am also a DRL addict, I would be curious to train one of the models with evolutionary algorithms (CMA-ES over parameters, for instance) to assess whether RL has an impact on compositionality (or it is solely the experimental protocol that matters).
 - I may have missed this point, but how many seeds did you use to run your experiments?
 - I may have also missed this point, what is the average length of the dialogue. Can you upload (non-understandable) dialogue example?

Last point... but it does not undermine the soundness of the experimental protocol!
 - In the end, Is 25 accuracy points really compositionality? What would be the score of simple strategies with overcomplete tokens? What is the score of the minimal vocab if we are only correct with one modality, two modalities?



Remarks:
 - in the introduction, you mention that previous old agents have grounded language, I am not sure whether we can speak of grounded language here, they have a predefined language, but it is not grounded.
 - Please remove the bold sentence in the introduction :) The claim is clear!
 - P11: Alg undefined
 - P12: the legend cannot be read


Conclusion
I am familiar with this type of experimental protocols, and I am well aware that they are never-ending works. There are always more experiments to do, more parameters to analyze. The final question is the following: is this paper have enough of these never-ending experiments? I think that this paper is just above this threshold by a short margin, and I vouch for weak accept.

However, I am missing at least one dynamic figure (to see the impact of the population along time, which is one of the core concepts of the paper), and there are several links with other ML communities that still have to be highlighted (especially evolutionary algorithms).
Besides, I somehow feel that the authors pursue two different goals in this paper: they both analyze memory/memoryless complete/overcomplete agents, which is somehow orthogonal to the general transmission hypothesis. Maybe, It would have made more sense to focus on one (or two) of the models and change the experimental setting on them (population size, training time, etc.)

In the end, I would favor a weak accept.
I am open to discussion regarding this scoring.

[1] Bäck, Thomas, and Frank Hoffmeister. "Extended selection mechanisms in genetic algorithms." (1991).
[2] Lanctot, Marc, et al. "A unified game-theoretic approach to multiagent reinforcement learning." Advances in Neural Information Processing Systems. 2017.
[3] Hinton, Geoffrey, Oriol Vinyals, and Jeff Dean. "Distilling the knowledge in a neural network." arXiv preprint arXiv:1503.02531 (2015).

**Experience Assessment:**

I have published one or two papers in this area.

**Review Assessment: Checking Correctness Of Derivations And Theory:**

N/A

**Review Assessment: Checking Correctness Of Experiments:**

I carefully checked the experiments.

**Review Assessment: Thoroughness In Paper Reading:**

I read the paper at least twice and used my best judgement in assessing the paper.

---

> ### Author Response · Authors · 2019-11-14
> **Responses to R3**
>
> Thanks for the well balanced review and depth of feedback! We generally agree with the comments and suggestions and add clarifications, additional information, and some rebuttal below.
>
> __Comment (referencing Kottur et al)__: “The authors made were careful not to take ownership of Kottur et al. 's works. Yet, the writing sometimes gives the feeling that their work is solely an extension of Kottur's work, which is not the case.”
> __Response__: Thanks! This is something we struggled with, so it’s nice to get feedback. We will try to better isolate references and clearly indicate the differences.
>
> __Comment (population dynamics)__: “Population dynamics: I am missing a key element in the paper: an analysis of the population dynamics. Although the paper deals with generational transmissions, there are no experiments that analyze the evolution of language generations after generations. Most of the experiments deal with the final convergence state.”
> __Response__: In initial experiments we measured D (the language dissimilarity metric from section 5.2) over training iterations. We will compute this plot for our final experiments and report it in the appendix. Initial experiments looked as expected: After some initial stabilization D looks like a typical learning curve for the 3 Multi Agent replacement methods, decreasing quickly at first then continuing to decrease slowly in later generations. The No Replacement strategy converges immediately and then either stays fixed throughout training or sometimes (for Small Vocab models) actually increases over the course of training.
>
> __Comment (literature)__: “Literature side: The authors did an excellent job [...] I think that it is worth extending[...]”
> __Response__: Thanks for pointing out the relations to game theory and dynamic distillation. Previous drafts did briefly discuss the relation to evolutionary algorithms, but we ended up cutting that discussion to help meet the page limit. We will add these changes as space allows.
>
> __Commen (qualitative figure clarification)__: “I had some difficulties in understanding Fig4, and the final-take away correctly. Would it be possible to give me one or two examples to correctly parse the table? More generally, I would recommend[...]”
> __Response__: Thanks for the suggestion. We will add further explanation along the lines of what follows to the paper: Consider the top left example of Fig 4a. In this case A-bot is presented with a dashed blue circle. Due to space constraints, the figure leaves out what Q-bot says. A-bot’s first response is the symbol “2” and A-bot’s 2nd response is the symbol “0”. The checkmark indicates Q-bot guessed correctly. The solid green star is a shape in the test set (because the text is in bold). After the first generation (Fig 4a) Q-bot guessed some other shape. After subsequent generations (Figs 4b and 4c) Q-bot guessed it correctly, qualitatively demonstrating the improvement in test accuracy we see after many generations. We will clarify our explanation in the final version.
>
> __Comment (meaning of D)__: “In a similar spirit, it is hard to interpret the distance in Figure 3.”
> __Response__: We only intended this metric to be used to compare models and we find it hard to ground in an absolute sense. The Single Agents Combined (roughly most different language) and Random Initialization (roughly most similar language) baselines can be compared to to get a sense of the range of performance for a particular model.
>
> __Comment (raw results table)__: “reproducible: having a final table in the array in the appendix could be very helpful”
> __Response__: We will add this.
>
> __Comment__: Number of seeds and average length.
> __Response__: Dialogs all take 2 rounds and we use 4 different random seeds. This makes 16  runs per experiment because there are also 4 splits. (As we mention in sections 2 and 4)
>
> __Comment__: “Last point... but it does not undermine the soundness of the experimental protocol! In the end, Is 25 accuracy points really compositionality?[...]”
> __Response__: Our models are usually not completely compositional in the sense of 100% test accuracy, but we seem to agree that an intermediate sense of compositionality (between 0% and 100% test accuracy) is useful. What simple strategies should we consider in addition those we already reported? With respect to modalities, is the reviewer asking about single attribute accuracy (One as in the One vs Both accuracy from Kottur et al)?
>
> __Comment (grounding)__ “[...]I am not sure whether we can speak of grounded language here[...]”
> __Response__: The observations made by A-bot are of simple attribute tokens fed through a learned embedding. They are not perceptual observations, so the language is not grounded in any perceptual environment in the normal sense. We think it is reasonable to say the language is grounded in these tokens, but agree the perspective is more controversial than the normal sense of grounding. We will make this discussion a bit more delicate.

---

> > ### Comment · AnonReviewer3 · 2019-11-15
> > **Response**
> >
> > Thank you very much for these clarifications. They are clear and I greatly appreciated that the authors expect to add temporal figures.
> >
> > Two concerned were not fully answered:
> >  - why 8 generations (and the potential discussions/experiments that may follow). Maybe the temporal plots would answer those questions, but they were not uploaded.
> >  - I cannot help myself to think that there are two directions in this paper. Transmission and Memory agents. Therefore, the overall message may be not as impactful as it should be (even if the claim is sound and the experiments seems sound).
> >
> > In the end, I am willing to increase my score from weak accept (6/10) to 7/10 (even if it does really exist :-/) but I cannot vouch for a clear accept (8/10).

---

> > > ### Author Response · Authors · 2019-11-15
> > > **Responses to R3**
> > >
> > > __Why 8 generations:__
> > > We agree that additional temporal analysis would strengthen the paper. In our initial experiments we focused on the Overcomplete setting and found that agent populations tended to converge by 8 generations. We've now performed some additional experiments that analyze test accuracy over time, though these are only preliminary due to other pending deadlines (CVPR). See [here](https://pasteboard.co/IGPZzwY.png) for a plot of test accuracy vs q-bot generation. See [here](https://pasteboard.co/IGQ0iZQ.png) for a plot of test accuracy vs a-bot generation. In both cases accuracy increases over generations, especially for the models that do better in the end. This more recent analysis suggests performance may improve for more generations for some models.
> > >
> > > __Memory:__
> > > The agent memory settings are a point of comparison with prior work (sorry for citing again but Kottur et al.), where they were shown to be useful in inducing more compositional language. As we also duplicate the work of Kottur et al. and introduce a more rigorous evaluation (both in terms of the compositional split and computing variance over multiple runs), we felt these settings were useful to the community for comparison. We note that our findings are consistent with this past work and that our approach is shown to be complementary. In updated drafts we will try to make this point clearer or deemphasize these results -- as the reviewer said Kottur et al. is not the only valid experimental setting.
> > >
> > > Again, thank you for your active participation in the review process!

---

### Official Review · AnonReviewer4 · 2019-10-28
**Official Blind Review #4**

**Rating:** 6

**Review:**

This paper proposes a simple extension to the training of emergent communication protocols in multi-agent settings.
The central hypothesis is that learnability will favor more 'compressible' and therefor more compositional languages to emerge.

This hypothesis is tested by training a population of listens and speakers in an emergent communication task and comparing a number of different strategies for reinitializing agents in the population. Since the new agents start from a random initialization, they provide a learning signal that reinforces protocols which can be learned quickly.

Experiments:
While the experimental results largely confirm the hypothesis there are a few issues:
-all of the plots show mean and standard deviations, rather than error of the mean. This makes it difficult to understand which differences are statistically significant and which ones are not.
- The replacement strategy 'epsilon-greedy' replaces agents based on their validation loss, which seems like an unfair advantage.
- It is currently unclear how much we can learn from Section 5.2: Naturally, agents trained in the same population will develop more similar protocols than those trained independently. This result is obvious and it is unclear whether reinitializing the agents makes any significant difference to the similarity. Again, confidence intervals would help.
- The experiments are also extremely toy. I would be more convinced if the authors tested their method on a more challenging task, though I am aware this is a common problem in this field.

Novelty:
The single biggest issues with the current form of the paper is the related work section. In this section two papers [1,2] are mentioned as "concurrent" when at least one of them [1] has been available online since June 2019. I think it is important to clearly point out the novelty of the current work compared to those two previous papers.
In particular [1] seems to be extremely close to the ideas and methods proposed here. Saying that these papers "confirm the hypothesis" simply is not enough.

References:
[1]: "Ease-of-Teaching and Language Structure from Emergent Communication", Funshan Li et al
[2]: "Co-evolution of language and agents in referential games", Gautier Dagan et al

[Updated score based on the rebuttal]


**Experience Assessment:**

I have published in this field for several years.

**Review Assessment: Checking Correctness Of Derivations And Theory:**

N/A

**Review Assessment: Checking Correctness Of Experiments:**

I assessed the sensibility of the experiments.

**Review Assessment: Thoroughness In Paper Reading:**

N/A

---

> ### Author Response · Authors · 2019-11-14
> **Responses to R4**
>
> Thanks for the review! We hope these responses satisfactorily address the raised concerns, especially those about novelty.
>
> __Comment (novelty)__: “The single biggest issues with the current form of the paper is the related work section. In this section two papers [1,2] are mentioned as "concurrent" when at least one of them [1] has been available online since June 2019. I think it is important to clearly point out the novelty of the current work compared to those two previous papers. In particular [1] seems to be extremely close to the ideas and methods proposed here. Saying that these papers "confirm the hypothesis" simply is not enough.”
> __Response__: Actually, our work has been available online since April 2019. Since this is prior to both [1] and [2], we do think it is fair to call our work “concurrent” and we do not think this is a valid reason for rejection. Neither of these works have been published yet, though [1] will appear at NeurIPS 2019. Furthermore, [2] builds on and compares to our work though we urge reviewers to avoid searching out identifying details.
>
> __Comment (statistical significance)__: “all of the plots show mean and standard deviations, rather than error of the mean. This makes it difficult to understand which differences are statistically significant and which ones are not.”
> __Response__: We take the reviewer’s point and will update our paper with standard error instead of standard deviation. However, we also point out that we tested the statistical significance of our results by applying t-tests to each pair of experiments. We find that most differences in performance are quite significant (p much less than 0.05). We use these results to support our discussion throughout section 5.1 and report all pairwise p-values in section A.3 of the appendix.
>
> __Comment (epsilon greedy)__: “The replacement strategy 'epsilon-greedy' replaces agents based on their validation loss, which seems like an unfair advantage.”
> __Response__: Absolutely. We agree this gives the Epsilon Greedy replacement strategy an advantage over the others because it is the only one that has access to validation performance. However, this is not an unrealistic advantage. Our agents can be given access to validation data as is done throughout the Meta Learning literature. Indeed, an interesting result of our experiments is that Epsilon Greedy did not help more.
>
> __Comment (language similarity)__: “It is currently unclear how much we can learn from Section 5.2: Naturally, agents trained in the same population will develop more similar protocols than those trained independently. This result is obvious and it is unclear whether reinitializing the agents makes any significant difference to the similarity. Again, confidence intervals would help.”
> __Response (obvious)__: We also were not very surprised that the Multi Agent settings had more similar language than the Single Agents Combined setting. We reported the result mainly to help orient the reader’s interpretation of the metric D and partially out of a desire to verify our intuition.
> __Response (significance)__: We will report standard error instead of standard deviation in Figure 3. We have computed p values and found that all pairs of No Replacement methods with Replacement methods have quite low t-test p values (much less than 0.05). This suggests the differences are significant. We will report these p values in the appendix.

---

> > ### Comment · AnonReviewer4 · 2019-11-15
> > **Response**
> >
> > These comments do indeed address my major concerns. I would have preferred for the paper to be already updated with the new plots but will increase my score based on the response, in particular regarding the novelty of the result and the significance tests.

---

### Decision · Program_Chairs · 2019-12-19

**Decision:**

Reject

**Comment:**

This paper explores the emergence of language in environments that demand agents communicate, focusing on the compositionality of language, and the cultural transmission of language.

Reviewer 1 has several suggestions about new experiments that are possible. The AC does think there is value in many of the suggested experiments, if not to run, then just to acknowledge their possibility and leave for future work. The reviewers also point to some previous work that is very similar.  E.g. "Ease-of-Teaching and Language Structure from Emergent Communication", Funshan Li et al